# A Computational Analysis Based on Automatic Digitization of Movement Tracks Reveals the Altered Diurnal Behavior of the Western Flower Thrips, *Frankliniella occidentalis*, Suppressed in *PKG* Expression

**DOI:** 10.3390/insects16030320

**Published:** 2025-03-19

**Authors:** Chunlei Xia, Gahyeon Jin, Falguni Khan, Hye-Won Kim, Yong-Hyeok Jang, Nam Jung, Yonggyun Kim, Tae-Soo Chon

**Affiliations:** 1Research and Development, Ecology and Future Research Institute (EnFRI), Busan 46241, Republic of Korea; c.xia2009@gmail.com (C.X.); hw102800@daum.net (H.-W.K.); janegsa1@gmail.com (Y.-H.J.); uchprecacia@gmail.com (N.J.); 2Research Institute of Computer, Information and Communication, Pusan National University, Busan 46241, Republic of Korea; 3Department of Plant Medicals, Andong National University, Andong 36729, Republic of Korea; gsh07129@gib.re.kr (G.J.); falgunikhan2942@gmail.com (F.K.); 4Department of Electrical and Electronics Engineering, Pusan National University, Busan 46241, Republic of Korea

**Keywords:** diurnal behavior, neural network, thrips, PKG, RNA interference

## Abstract

Diurnal behavior is a characteristic of the western flower thrips, *Frankliniella occidentalis*. It is controlled by the circadian clock machinery of the thrips. However, it was unclear how the clock machinery controls diurnal behaviors such as feeding or mating. This study hypothesized that a foraging gene encoding cGMP-dependent protein kinase (*PKG*) mediates diurnal behaviors. RNA interference with *PKG* expression led to significant alteration of clock gene expression and caused adverse effects on early-life development and adult fecundity. To examine behavioral alterations, adult movement was continuously observed for a 24 h period with an automatic digitization device. Diel difference was observed with speed and durations (%) in RNAi-control females but not in the RNAi-treated females. Three sequential stages consisting of high activity followed by feeding and visiting of micro-areas were observed for the control females but not in the RNAi-treated females. These results suggest that PKG mediates the diurnal behavior of *F. occidentalis* under the control of the circadian clock machinery.

## 1. Introduction

The western flower thrips, *Frankliniella occidentalis*, is polyphagous and causes damage to high-value crops including hot peppers [1]. The insect pest also transmits plant viruses to crops, causing devastating economic damage [2]. Due to frequent use of chemical insecticides, this species has developed insecticide resistance [3]. Moreover, its ability to hide in flowers or crevices makes it difficult to effectively control by insecticide sprays [4]. Alternatively, non-chemical control techniques have been developed and implemented to suppress the outbreaks of *F. occidentalis*. For example, aggregation pheromone has been used in sticky traps for mass-trapping [5,6]. In addition to chemical cues, visual signals during day time are used to locate hosts and facilitate diurnal feeding and mating rhythmicity [7,8]. Four circadian clock genes (Period (*Per*), Timeless (*TIM*), Doubletime (*DBT*), and Clock (*CLK*)) are expressed in *F. occidentalis* and associated with the diel rhythmicity [8].

Insect feeding behavior is functionally associated with expression of a foraging gene encoding a cGMP-dependent protein kinase (*PKG*) [9]. Upon activation by cGMP, PKG phosphorylates a number of biologically important targets associated with the regulation of muscle contraction, metabolism, and gene expression [10]. Variation of *PKG* expression is implicated in the plasticity of insect behaviors [11]. Manipulation of its expression level and subsequent kinetic activity influences alternative feeding behaviors of the fruit fly, *Drosophila melanogaster;* the two patterns are referred to as sitters and rovers [12]. The expression levels also drive division of labor systems across diverse social species [13,14]. These suggest a functional role of PKG in controlling insect behaviors. In mammals, the circadian clock is entrained by signals mediated through PKG activity, which is induced by cGMP up-regulated by nitric oxide (NO) which catalyzes the phosphorylation of TIM [15]. This suggests that PKG influences the circadian clock of *F. occidentalis*. However, the functional link between PKG and circadian clock function remained elusive in insects.

This study investigated the physiological role of PKG in the diurnal behavior of the thrips, *F. occidentalis*, by regulating expression of the clock genes. To address this hypothesis, PKG gene expression in *F. occidentalis* was monitored over a 24-h period in a 12:12 LD cycle. To test the functional link, *PKG* expression was suppressed by its specific RNA interference (RNAi) and the resulting changes of the clock gene expressions were analyzed. In addition, any subtle alteration of the thrips behavior was assessed by mathematical parameters extracted from continuous automated 24-h monitoring of movement tracks.

## 2. Materials and Methods

### 2.1. Insect Rearing

Both larvae and adults of *F. occidentalis* were obtained from the Department of Crop Protection, National Institute of Agricultural Sciences (Jeonju, Republic of Korea), and maintained at conditions of 25 ± 1 °C temperature, 60 ± 5% relative humidity, and a 14:10 h (LD) light cycle. Newly germinated beans (*Phaseolus coccineus* L.) were supplied for feeding and oviposition. Eggs that were newly laid on the beans in adult colonies were transferred to a breeding dish (SPL Life Science, Seoul, Republic of Korea). After 3 days, by which point most larvae had hatched, new beans were supplied every day. Under the laboratory conditions, larvae underwent two instars (L1–L2) and were distinct from prepupae or pupae that developed wing pads.

### 2.2. Bioinformatics Analysis

DNA and amino acid sequences including the *PKG* gene of *F. occidentalis* (*Fo-PKG*) were obtained from NCBI (National Center for Biotechnology Information: https://blast.ncbi.nlm.nih.gov, accessed on 1 October 2024) with accession numbers. MEGA6.0 was used to construct a phylogenetic tree through clustering using Maximum likelihood, where the evolutionary distances were computed using the Poisson correction method. Bootstrapping values were obtained with 1000 trials to support branching and clustering. Protein domains were predicted using a program searching the conserved domain (https://www.ncbi.nlm.nih.gov/Structure/cdd/wrpsb.cgi accessed on 1 October 2024), and InterPro (https://www.ebi.ac.uk/interpro/ accessed on 1 October 2024). The N-terminal signal peptide was determined using the SignalP 5.0 server (https://services.healthtech.dtu.dk/service.SignalP-5.0/ accessed on 1 October 2024). The resulting domains were drawn by Biorender (https://biorender.com/ accessed on 1 October 2024). The protein–protein interaction map was generated by using STRING 12.0 (https://string-db.org accessed on 1 October 2024).

### 2.3. RNA Extraction and cDNA Preparation

Total RNAs were extracted using Trizol reagent (Invitrogen, Carlsbad, CA, USA) according to the manufacturer’s protocol. For each RNA extraction, 25 females were macerated using 500 μL of Trizol reagent. Following the RNA extraction, RNA was resuspended in 30 μL of diethylpyrocarbonate (DEPC)-treated water and quantified using a spectrophotometer (Nanodrop, Thermo Fisher Scientific, Wilmington, DE, USA). For cDNA synthesis, 400 ng of RNA was used in each sample with RT oligo dT premix (Intron Biotechnology, Seoul, Republic of Korea) containing oligo dT primer. A reaction mixture consisted of 2 μL of RNA extract and 18 μL of DEPC-treated water and was run according to the manufacturer’s instructions. The resulting cDNA samples were kept at −20 °C before being used for experimentation.

### 2.4. RT-PCR and RT-qPCR

RT-PCR used the cDNA and amplified *Fo-PKG* and two clock genes with a Taq polymerase (GeneALL, Seoul, Republic of Korea). A reaction mixture for PCR consisted of 2.5 μL of dNTP (each 10 pmol), 2.5 μL of 10× Taq buffer, 2 μL of forward and reverse primers (10 pmol/μL, Appendix A), 0.5 μL of Taq polymerase, 1 μL of cDNA, and 16.5 μL of distilled deionized water. The PCR conditions began with an initial denaturation at 95 °C for 5 min, which was followed by 35 amplification cycles consisting of 95 °C for 1 min, 50~55 °C for 30 sec, and 72 °C for 1 min. At the end of the amplification cycle, an additional extension was performed at 72 °C for 5 min. The PCR product was confirmed by 1% agarose gel electrophoresis. The qPCR used a Step One Plus Real-Time PCR System (Applied Biosystem, Waltham, MA, USA) under the guidelines of Bustin et al. [16]. A sample of qPCR reaction (20 µL) contained 10 µL of Power SYBR Green PCR Master Mix (Toyobo, Osaka, Japan), 3 µL of cDNA template (100 ng), and 1 µL (10 pmol) each of the forward and reverse primers (Appendix A). After an initial heat treatment at 95 °C for 2 min, qPCR was performed with 40 cycles of denaturation at 95 °C for 30 s, annealing at 50−55 °C for 30 s, and extension at 72 °C for 30 s. The transcript levels of elongation factor 1 (EF1) were used as a reference for the normalization of each test sample. Quantitative analysis was conducted using the comparative CT (2^−ΔΔCT^) method [17]. All experiments were independently replicated three times.

### 2.5. RNA Interference (RNAi) Treatment and Subsequent Behavior Assays

Template DNA was amplified with gene-specific primers (Appendix A) containing T7 promoter sequence (5′-TAATACGACTCACTATAGGGAGA-3′) at the 5′ end. The PCR conditions were as described above. After confirming the PCR product, the resulting PCR product was used to synthesize double-stranded RNA (dsRNA) encoding Fo-PKG using T7 RNA polymerase with NTP mixture at 37 °C for 3 h (MEGA script RNAi kit, Ambion, Austin, TX, USA). dsRNA (2 μg/10 μL) specific to *Fo-PKG* (‘dsPKG’) was mixed with a transfection reagent Metafectene PRO (Biontex, Plannegg, Germany) at a 1:1 (*v*/*v*) ratio and incubated at 25 °C for 30 min to form liposomes, and the resulting mixture was supplied with beans to *F. occidentalis*. Beans were coated with the dsRNA mixture of 10 μL per 20 thrips and fed for 12 h. Control dsRNA (‘dsCON’) was prepared according to the method outlined by Vatanparast et al. [18].

### 2.6. Fluorescence In Situ Hybridization (FISH) Assay

Adult thrips were tested using FISH assay to detect the expression pattern of *Fo-PKG*. The guts of thrips were dissected upon a sterilized glass slide and then treated with 4% paraformaldehyde for 1 h at room temperature. The guts were permeabilized with 1% Triton X-100 in PBS for 2 h at room temperature after being cleaned with 1× PBS. After that, the gut was washed with 1× PBS, rinsed with 2× SSC, and incubated for 1 h at 42 °C in a dark, humid environment with 25 μL of pre-hybridization solution (2 μL of yeast tRNA, 2 μL of 20× SSC, 4 μL of dextran sulfate, 2.5 μL of 10% SDS, and 14.5 μL of deionized distilled water). The buffer was then replaced with hybridization buffer (5 μL of deionized formamide and 1 μL of oligonucleotide (10 pmol) tagged with fluorescein in 19 μL of the pre-hybridization buffer). The DNA oligonucleotide probes were tagged with fluorescein amidite (FAM) at the 5′ end and purified using high-performance liquid chromatography at Bioneer (Daejeon, Korea). To identify *Fo-PKG* mRNA, an antisense probe (5′-FAM-TTT-TCAGTCACATTGTGTACGTT-3′) that is complementary to the target mRNA and a negative dsCON sense probe (5′-FAM-AAA-AACGTACACAATGTGACTGA-3′) were produced at a concentration of 10 pmol. The slides were then sealed with an RNase-free coverslip and left in a humid room at 42 °C for 16~18 h. Following hybridization, the gut was incubated with 4× SSC containing 1% Triton X-100 at room temperature for 5 min, and it was rinsed twice with 4× SSC for 10 min each. The gut samples were washed three times with 4× SSC and then incubated at 37 °C with a 1% solution of anti-rabbit-FITC conjugated antibody (Thermo Fisher Scientific) in PBS for 30 min under darkness. The guts were incubated with 4x SSC for 10 min, and then with 2× SSC for 10 min. The samples were viewed under a fluorescence microscope (DM2500, Leica, Wetzlar, Germany) at 200× magnification after adding a drop of PBS/glycerol (1:1, *v*/*v*) and incubating at room temperature for 15 min.

### 2.7. Behavior Monitoring and Automatic Digitization

Individual adult female thrips at 2–3 days after emergence were observed continuously for 24 h in each trial. The detailed observation system is provided in Appendix A. The observation system was installed on an optic fiber microscope (Camscope^®^, AST-ICS305B, Sometech Vision, Seoul, Republic of Korea) consisting of observation arena, camera, computer, and light (Appendix A). The observation arena (6 mm in diameter and 1.7 mm in depth) was made of dental modelling wax [19] and was divided into food-provision (2.5 mm radius), intermediate (0.5 mm width) and edge (1 mm width) areas (Appendix A). According to measurements of 20 adult females, the average length, width and height were 1.64 ± 0.07 mm, 0.29 ± 0.03 mm and 0.25 ± 0.01 mm, respectively. According to preliminary experiments, the tested females moved around over distances about three times of either the width or height of their bodies along the edge area; 1 mm was therefore determined as the width of the edge area. The intermediate area was defined as the area between the food-provision area and the edge area. The parameters were measured in the whole observation arena, while durations (%) were separately measured for different micro-areas including the food-provision, intermediate and edge areas (See Appendix A). Within the food-provision area, a particle (2 mm in diameter and 1 mm in height) of fresh bean just after germination was provided as food in the center. Water was provided for the food continuously to keep the food fresh during the observation period (Appendix A).

The temperature was 23.6 ± 2.0 °C and humidity was 55.3 ± 10.0% in the observation room. Photophase (14 L) and scotophase (10D) were provided using white and red LEDs, respectively. In order not to cause compound effects between transplant of test females and light phase change, the test females were introduced to the observation arena 2 h before the start of photophase for acclimation. Digital observation was conducted continuously for 24 h just after light-on in the light cycle.

Behavioral tracks were recorded with an image resolution of 1920 × 1080 with 15.88 frames per second (fps). Movement was recognized by a convolutional neural network, YOLOv8 [20] continuously in photo- and scotophases as examples shown in Appendix A. One second was defined as the observation time unit to obtain movement parameters and duration rates in percentage (durations (%)) in this study, although time intervals of less than 1 s (e.g., 0.25 s) have been used in similar research to monitor the movement of insects (e.g., *Drosophila melanogaster*) responding to external stimuli (toxins) [21,22,23]. Measuring the parameters at a time interval of 1 s was sufficient to present the movement and positioning status over the entire 24 h period. The time unit in 1 s also reduced the computational time.

Speed, locomotory rate, and direction change rate (DCR) were recorded for movement parameters, while durations (%) in different micro-areas were obtained to indicate where test animals stayed within the observation arena as time progressed. Whereas speed was obtained over the entire observation period including the time without movement, locomotory rate was defined as the mean speed only when the test individuals moved. DCR was calculated as the angle change (without considering direction) after one unit time (1 s). Duration of staying at the food-provision (edge) area was measured as the total period while the digitized body center was located either within or on the border of food-provision (edge) area. The rest of the period was regarded as duration in the intermediate area. The parameters and durations (%) were measured in each time unit first, and averaged in each hour defined as a light phase (i.e., P1-P14 for photophase, and S1-S10 for scotophase).

### 2.8. Statistical Analysis

Percent data for genetic analyses were arcsine-transformed, and the subsequent transformed data were confirmed to follow a normal distribution using PROC UNIVARIATE of the SAS program [24]. Data obtained from the feeding or mating test were subjected to a one-way analysis of variance (ANOVA) using PROC GLM of the SAS program. The means were compared using the least significant difference (LSD) test at a Type I error of 0.05.

For behavioral monitoring, the mean of each trial (four trials for dsCON and three trials for dsPKG) was measured initially. Subsequently, means and standard deviations (SDs) of the trial means were obtained again as representative values of parameters and durations (%), according to the central limit theorem [25,26]. Two aspects of statistical difference were considered in this study: difference between dsCON and dsPKG at each light phase, and difference among light phases within each strain. Behavioral data in this study had a property of measurement dependence. Since observations were continuously conducted throughout the whole observation period, movement across light phases was dependently observed (i.e., observed continuously). In this study, measurements in different light phases were considered as independent because this is the initial phase of a behavioral study to confirm physiological effects, and an extensive experimental design would be required for analyzing repeated measurements, possibly with a large number of trials.

For analyzing multiple comparisons among light phases, each pair was selected separately and compared with each other, assuming measurements were independent. Considering high variability in behavioral data, the Mann–Whitney U test (or Wilcoxon–Mann–Whitney test) [27] was adopted for analyzing nonparametric data (rank) while the Kolmogorov–Smirnov test [28] was additionally adopted for analyzing parametric data (mean) for two-sample tests (four and three trials for dsCON and dsPKG, respectively, in each light phase). The Kolmogorov–Smirnov test could be effectively applied to the data not following the Gaussian distribution [28].

The software was obtained from ‘scipy.stats’, a Python package (V1.14.1). By combining two test results, a statistical differentiation score was devised to indicate a possibility of quantitative separation between photo- and scotophases. Scores 1 and 2 were given to probability of alpha error less than 0.10 and 0.05 for each test, respectively. Then, the scores for the two tests were summed to present overall statistical differentiation with a maximum of 4 and minimum of 1. In comparing statistical difference between dsCON and dsPKG in each light phase, the *t*-Test was applied to parameters and durations (%) between dsCON and dsPKG directly in each light phase (four and three trials for dsCON and dsPKG, respectively, in each light phase) with one-tail analysis under the condition of heteroscedasticity, considering difference in variance between two treatments.

## 3. Results

### 3.1. A PKG Gene Is Expressed in All Developmental Stages of F. occidentalis

An *Fo-PKG* gene (GenBank accession number: XM_026417160.2) is encoded in the genome of *F. occidentalis* (Figure 1). It encodes an open reading frame translated into a sequence of 694 amino acids, which is predicted to have two cAMP binding domains and a protein kinase domain (Figure 1A). A phylogenetic tree indicates that *Fo-PKG* is clustered with other insects’ *PKG* genes but away from the gene clusters of *PKA* and *PKC* (Figure 1B). *Fo-PKG* was expressed in all developmental stages but exhibited relatively low expression level in the pupal stage (Figure 1C). In young adults, both males and females had the highest expression levels of *Fo-PKG* at 3 days post-emergence (Figure 1D). The expression levels abruptly reduced at 5 days post-emergence.

### 3.2. Diel Rhythmicity of Fo-PKG Expression Profile

To understand the diel rhythmicity of *Fo-PKG* expression, its expression levels were assessed every 2 h in a day (Figure 2A). In the larval stage, two expression peaks occurred during photophase, at which points the overall gene expression levels were higher than those during scotophase (*F* = 95.10, df = 1, 10; *p* < 0.0001). The high expression pattern during photophase was maintained in the adult stage with a single high expression peak at 6 h after light-on. The overall expression levels during photophase were significantly (*F* = 82.43, df = 1, 10; *p* < 0.0001) higher than those during scotophase in the adult stage. The high expression of the PKG gene during photophase was visualized by FISH assay (Figure 2B). The fluorescence-labeled antisense probe specific to PKG mRNA specifically gave a positive signal in the midgut of the adults at 6 h after light-on, while its sense probe did not show any signal. However, the FISH signal was much weaker in the midgut during scotophase, at which time the normalized signal intensity was lower than that during photophase by almost two-fold.

### 3.3. Functional Link Between PKG and Clock Genes

To predict a functional link for the expression of *Fo-PKG*, its protein–protein interactions were assessed using a String database (Figure 3A). This analysis indicated that PKG is directly linked with NO signaling, as expected. Interestingly, PKG is linked to clock genes such as period (*Per*) and Clock (*CLK*) via casein kinase (*CK*) and cAMP-dependent protein kinase (*PKA*).

To investigate a functional association between PKG and the two clock genes, the loss-of-function approach was applied by RNAi specific to *Fo-PKG*. Feeding dsPKG significantly suppressed the *Fo-PKG* expression in larvae and adults (Figure 3B). The RNAi effects on the survival rate and reproduction rate of *F. occidentalis* were analyzed (Appendix A). In the case of total thrips fed with dsPKG for 24 h, the survival rate decreased by approximately 20% starting from the third day (Appendix A). Additionally, the oviposition rate showed a statistically significant decrease in the number of progenies laid per female when treated with dsPKG (Appendix A). Thus, at least for one day after RNAi treatment, the thrips did not show any detrimental influence in terms of survival.

Under these RNAi conditions, the expressions of *CLK* and *Per* genes were assessed at two time points in both photophase and scotophase (Figure 3C). The RNAi treatment against *Fo-PKG* expression significantly reduced the expressions of *CLK* and *Per* genes during photophase in both larvae and adult stages. However, the RNAi treatment did not influence the clock gene expression during scotophase.

### 3.4. Parameter Extraction and Behavior Profiles of Adult Females

From the digitized movement data, parameters including speed, locomotory rate and Direction change rate (DCR) were extracted throughout the whole observation period (1 day). Concurrently, durations (%) were measured in different micro-areas including the food-provision, intermediate and edge areas as time progressed (See Appendix A).

Speed and locomotory rate in the total period were higher for dsCON (0.13 ± 0.12 mm/s) than those for dsPKG (0.07 ± 0.07 mm/s) (Table 1). DCR was overall similar between the two treatments. Durations (%) were substantially higher in the edge area (70.0~73.3%) than the food-provision (13.3~18.4%) and intermediate (8.3~16.7%) areas in both treatments. It is noted that durations (%) in the intermediate area were higher for dsPKG (16.7%) than for dsCON (8.3%).

Diel difference was observed with speed in dsCON with higher levels in photophase (0.16 ± 0.13 mm/s) than in scotophase (0.08 ± 0.08 mm/s) (Table 2). Locomotory rate was almost the same as the speed in photo- and scotophases. Diel difference was not observed in DCR in both treatments. It is noted that durations (%) in the intermediate area were higher in photophase than in scotophase in both treatments with 11.9 and 3.4, respectively, in dsCON, and 19.6 and 12.6, respectively, in dsPKG. Durations (%) in photo- and scotophases in both treatments were overall alike, in the range 12.2~23.8 in the food-provision area and 66.3~75.2 in the edge area (Table 2).

Behavior profiles across light phases were presented with speed superimposed over durations (%) as time progressed (Figure 4). Behavior patterns were divided into three consecutive stages in dsCON females. The stage started from the middle of the observation day for convenience of comparison. The first stage included the changing phase from photophase to scotophase, P10~S5, with maximal durations (%) in the edge area and minimal durations (%) in both food-provision and intermediate areas. The speed had a peak (0.25 mm/s) during P12 (arrow, Figure 4A), decreasing linearly afterward. In this stage, the females were not active in feeding and mainly moved around in the edge area especially in the initial period. In the second stage, S6–S10, durations (%) in the food-provision area substantially increased to a maximal level (40.1 on average), indicating active feeding (Figure 4A). In the following third stage in photophase, P1~P9, durations (%) in the intermediate areas (15.3 on average) were high, especially in P3~P9, compared with the previous stages. Considering duration rates (%) in the food-provision (19.0 on average) and edge areas (65.7 on average) were also high, females moved around across different micro-areas actively in this stage, partly involved in feeding. It is noted that speed was very high in the early photophase, P1~P3, in this stage. This high speed may be due to the energy accumulated by feeding in the previous stage, S5~S10 (Figure 4A).

It is also noted that speed was differentiated according to micro-areas across light phases. The females mostly stayed in the edge area (87.1 ± 9.5%) during P10–P12. Speed was correspondingly very high (0.22 ± 0.10 mm/s) during this period with a peak during P12 (Figure 4A). Moreover, the speed decreased to a low level (0.09 ± 0.09 mm/s) afterward during P13–S6, while the duration in the edge area was still high with 85.8 ± 21.5%. This indicated that speed changed within the edge area as time progressed. In the early photophase, a decrease in speed was similarly observed. The speed was initially at its highest level of 0.31 ± 0.19 mm/s during P1 and rapidly decreased in the following light phases, P2, P3 and P4, with 0.22 ± 0.15 mm/s, 0.18 ± 0.18 mm/s and 0.08 ± 0.05 mm/s, respectively. In this period, the duration rates in the edge area also decreased, 84.1 ± 15.7%, 72.8 ± 25.4%, 66.9 ± 28.7%, 64.2 ± 27.5%, respectively (Figure 4A). During the period S6–S9, duration in the food provision area increased to 44.5 ± 40.8%, somewhat like duration in the edge area (51.5 ± 40.4%). During this period, speed was in the intermediate range, 0.08 ± 0.09 mm/s. The results overall indicated that speed was variable according to micro-areas as time progressed, especially in the edge area for dsCON.

Behavior profiles of the dsPKG (Figure 4B) were substantially different from those of dsCON. Overall, speed and durations (%) were irregular with intermittent peaks across light phases. In the initial stage, P10~S5, durations (%) in the intermediate area substantially increased (41.6 on average), especially during P12~S3. Three peaks of speed (0.10~0.13 mm/s) were also observed in this period. It is noted that durations (%) in the food-provision area were high at both beginning (P10~P12) and end (S2~S5) of this stage, indicating that the test females would be involved in feeding in these periods. Behaviors in the second stage in S6~S10 were also contrastingly different from the control. The test females stayed in the edge area (81.5% on average) for almost the whole period, rarely visiting the food-provision area (13.3% on average). Speed was in the intermediate range (0.05 mm/s on average) in this stage. In the third stage in photophase, P1~P9, females variably visited different micro-areas with intermittently high levels of speed between 0.08~0.18 mm/s. Speed was not characterized according to different micro-areas overall for dsPKG, since the parameters and durations (%) were irregular in this strain. Although irregular overall, it is noted that a sharp decrease in speed was observed from 0.18 mm/s to 0.03 mm/s along with the decrease in durations (%) in the edge area during P1~P3 (Figure 4B), like the case shown for dsCON (Figure 4A). Also, long durations (%) in the intermediate area were observed from P12 to S2 (38.7%~63.7%), while speed was in the middle range (0.03~0.10 mm/s) during this period.

### 3.5. Comparison of Movement Trends Between dsCON and dsPKG

The trends of movement parameters were compared between dsCON and dsPKG in detail in Figure 5. Diel difference in speed was observed with the dsCON females across light phases as stated above. The speed of the dsCON females was initially high at P1 (0.31 mm/s), remained at a low level (0.11 mm/s on average) during P5~P8, and increased again from P9 to reach a peak at P12 (0.25 mm/s) (Figure 5A). Afterward, speed decreased continuously until S1 and was low almost throughout the whole period of scotophase (0.08 mm/s on average). For dsPKG-treated females, however, this type of diel difference was not observed, with speed at consistently low levels (0.07 mm/s on average) throughout the observation period, except intermittently high levels between 0.13~0.18 mm/s. The trends of locomotory rate of dsCON and dsPKG were very similar to those of speed (Figure 5B), reflecting that the stops in movement did not much influence motility of test females measured with the time unit at 1 s. DCRs across light phases were overall stable in both dsCON and dsPKG with a relatively higher degree of fluctuation observed in ds-PKG (Figure 5C).

Statistical significance was further obtained for comparing parameters between dsCON and dsPKG in each light phase according to *t*-Test (See Section 2). A few light phases showed statistical differences. Significant differences in speed showed mainly in late photophase, P9–P11, between dsCON and dsPKG (Figure 5A). The light phases with alpha error up to 0.10 are also presented in the Figure in P2 and P12. Locomotory rate had similar significance to the case of speed (Figure 5B). DCR had statistical significance during P2 and P10 with an alpha error of 0.05 (Figure 5C).

Durations (%) staying in the food area (Figure 6A) were initially similar between the treatments with low levels from early photophase to early scotophase (P1~S2) (dsCON; 13.69% and dsPKG; 13.32%). Afterward, substantial differences between dsCON and dsPKG were observed in two phases. Early in S2 and S5, percentages of time in the food-provision area were higher with dsPKG (24.32% on average) than dsCON (8.26% on average). In the following light phases from S6 to S10, reverse durations were observed, with higher levels in dsCON (40.10% on average) and lower levels in dsPKG (4.72% on average).

In the intermediate area, durations (%) were characterized by an exceedingly high level for dsPKG during P12~S2 (Figure 6B). Except for this, durations (%) were overall stable for dsCON and dsPKG at low levels. It is noted that SDs were exceptionally high with durations (%) for dsPKG in P12~S2 (Figure 6B).

Percentages of staying in the edge area (Figure 6C) were overall similar in early-to-mid photophase, P1~P7, in dsCON (65.3% on average) and dsPKG (75.8% on average), although some levels of fluctuation were observed in both treatments during this phase. Subsequently, substantial differences were observed between dsCON and dsPKG in two phases. Early in P12~S5, durations (%) were lower for dsPKG (53.00% on average) than for dsCON (89.75% on average). Afterward, during S6~S10, durations (%) were reversed, being lower for dsCON (55.76% on average) than for dsPKG (90.36% on average). It is noted that the switching time for durations (%) was the same with S6 in both food-provision and edge areas (Figure 6A,C).

Durations (%) in different micro-areas between dsCON and dsPKG showed weak statistical significance compared with movement parameters (Figure 6). In the intermediate area, statistical significance was observed in one light phase, P6 (Figure 6B). The probability of alpha error up to 0.10 was only observed intermittently during a few light phases in the food-provision and edge areas (Figure 6A,C).

As shown in Figure 7, statistical differentiation scores in parameters and durations (%) among light phases were observed according to Mann-Whitney U and Kolmogorov-Smirnov tests. For dsCON, statistical significance in speed and locomotory rate with S4 vs. most photophase was noted with maximum scores equal to 4 (green arrows). Low degrees of differentiation with score 1 were also observed intermittently. In DCR, no meaningful statistical significance was observed. For dsPKG, speed and locomotory rate showed no statistical difference except for a few cases of intermittent differences with scores equal to 2, showing only one test of statistical significance in these cases. In DCR, no difference was observed overall except one case (P1 vs. P4) with the differentiation score equal to 2 in dsPKG treatment. Regarding durations (%) in the micro-areas, significances were also found with dsCON, more strongly in the intermediate area, while significances were not observed with dsPKG in all micro-areas (Figure 7). Like the cases of movement parameters, statistical differentiation of duration (%) was high in S4 vs. most photophase with the scores equal to 4 with dsCON in all micro-areas (green arrows). Also in the intermediate area, maximum statistical differentiations with score equal to 4 were found intermittently in P2, P5, S1, S5 and S7, along with minor differences with scores equal to 1. Intermittent statistical differentiations with score equal to 4 were found less in the food-provision and edge areas in photophase.

## 4. Discussion

The continuous monitoring of the thrips behavior showed a diurnal rhythmicity with relatively high activity during photophase and relatively low activity during scotophase. This kind of diel pattern is consistent with circadian regulation mediated through two oscillatory loops: the Per/TIM oscillatory loop and the CRY loop [29,30]. In each of these loops, CLK/CYC acts as a transcriptional activator that promotes *Per* and *TIM*, or *CRY* transcription. The product proteins Per and TIM, or CRY are thought to provide negative feedback to inhibit the transcriptional activator. In *D. melanogaster*, *CRY* is expressed in specific clock neurons in the brain [31]. CRY is then activated by blue light and catalyzes TIM degradation through its protease activity, at which point the circadian clock is reset [32,33]. In *F. occidentalis*, *Per* and *CLK* genes exhibited diel patterns with relatively high expressions during photophase and relatively low expressions during scotophase. This supports the circadian rhythmicity of thrips behavior controlled by clock genes, because any interruption of the clock gene expressions significantly altered the diel rhythmicity [8]. Interestingly, this current study showed that *PKG* expression followed this diel pattern, suggesting a functional association of its expression with clock gene expression. The expression pattern was visualized in the intestine by FISH. Even though the circadian clock is mainly controlled by the brain, the peripheral tissues should be adjusted to the rhythm. This study focused on feeding behavior to assess the circadian rhythm. The diel pattern of feeding behavior is controlled by central (brain) and peripheral (fat body and gut) signals because any mismatch between these signals would lead to significant decrease of feeding activity in *Drosophila* [34]. This supports the diel patterning of PKG expression in the intestine of thrips.

RNAi specific to *PKG* expression led to alterations in the clock gene expressions during photophase by preventing up-regulation of the clock genes. *PKG* expression is associated with division of labor in the honey bee, *Apis mellifera*, in which specific expression in the mushroom bodies of the brain, subesophageal ganglion, and corpora allata is associated with foraging behavior to collect pollens and nectars [9]. The role of *PKG* expression in labor division in social insects was also found in a fire ant, *Solenopsis invicta*, in which RNAi specific to *PKG* reduced the locomotory activity and facilitated the behavioral change from foragers to nurses [35]. PKG activity and locomotory activity has been well established in *Drosophila*, in which flies are discriminated into sitters with low PKG activity or rovers with high PKG activity [36]. These suggest that *PKG* expression is associated with high locomotory activities of *F. occidentalis* during photophase. Thus, the RNAi specific to *PKG* expression resulted in reduced early-life development and adult fecundity, presumably by inhibiting feeding and ovipositional behavior (Appendix A). In addition, the suppression of *PKG* expression led to the suppression of the clock gene expressions during photophase, supporting the functional association of *PKG* expression with circadian rhythmicity. In mammals, the suppression of clock gene expressions led to the suppression of PKG protein level [37]. This suggests that alterations of the clock gene expressions indirectly influence the reduced *PKG* mRNA level caused by RNAi in *F. occidentalis*. It is also noteworthy that there are at least four isoforms of PKG in the *F. occidentalis* genome: 176 (XP_052119469.1), 417 (XP_052132331.1), 694 (XP_026272945.1), and 1,010 (XP_026272942.1) amino acid residues. In this study, we analyzed only PKG with the 694 amino acid length isoform. Thus, the other isoforms should be analyzed in their independent roles. In addition, our current study showed altered expressions of two circadian genes by manipulating *PKG* expression, but did not show control of the *PKG* gene expression by circadian gene expression. Thus, it is still unclear whether the PKG gene’s effects on clock genes are causal or incidental with respect to the changes in behavior. This needs to be clarified in subsequent research.

Automatic individual recognition of live individuals and continuous parameter extraction demonstrated an impact on movement behavior of the thrips caused by PKG and associated clock genes. Diel rhythm in adult females with RNAi specific to *PKG* expression was substantially affected, without showing much variation in the activity parameters and durations of stay at different areas of the observation arena, whereas the dsCON females showed clear diel difference.

The continuous detection of behaviors over the entire period of 24 h effectively characterized diel difference between dsCON and dsPKG. Not only differences in movement parameters but also differences in durations (%) in the micro-areas in the observation arena were observed between dsCON and dsPKG (see Table 1). It is noted that total speed was high (0.13 mm/s) in dsCON compared with dsPKG (0.07 mm/s), indicating a higher level of energy consumption by dsCON.

Behavior profiles were produced in three stages by superimposing speed over durations (%) on continuously observed data (see Figure 4). Overall, behavior profiles were effectively characterized by sequential stages, activity followed by feeding and visiting of other micro-areas in dsCON-females, and corresponding behavior disruptions after dsPKG suppression (see Figure 4A,B). Currently, however, the causes of behavior profile changes are unknown. More studies are warranted regarding physiological and genetic aspects along with diverse experimental conditions in the future.

Although the means and SDs were highly variable in both movement parameters and durations (%), statistical significances according to Mann–Whitney U and Kolmogorov–Smirnov tests were found between photo- and scotophases in dsCON in both movement parameters and durations (%) in different micro-areas, whereas the difference was not observed in dsPKG (see Figure 7). It is noted that a period in scotophase, S4, mainly had statistical differentiations with most periods in photophase in both parameters and duration rates (%) in dsCON, confirming coincidence in movement parameters and durations (%) in behavioral changes. Phase S4, in mid-scotophase, had the minimum speed (0.19 mm/s) and locomotory rate (0.19 mm/s) along with minimal SD ranges (see Figure 5A,B). Similarly, durations (%) in S4 were either maximal in the edge area or minimal in the food-provision and intermediate areas, while SDs were commonly minimal in this phase (see Figure 6). The distinctive differences in mean values between light phases, concurrently with minimal range in SDs, contributed to presenting statistical significance originating from S4 in light phases.

Statistical significance of parameters was also examined specifically between dsCON and dsPKG in each light phase according to *t*-Test (See Section 2). The speed and locomotory rate showed significant differences mainly in three light phases, P9–P11, between dsCON and dsPKG (see Figure 5A,B). These differences were contrasted with the case of statistical differences within each treatment where S4 was different from photophase, as stated above (see Figure 7). The results suggested two notable areas of behavioral change: the late photophase showed greater differences when comparing dsCON and dsPKG effects in each light phase, while light phase S4 showed greater differences after dsPKG suppression in relation to other light phases. Since high variability existed in the behavioral data, results need to be confirmed with more investigations through integrative approaches linking behavior, physiology and genetics along with more trials.

It is noted that behavioral changes were more obvious in the intermediate area. Duration of stay in this area increased to 16.7% for dsPKG whilst remaining low, at 8.3%, for dsCON (see Table 1). In addition, the tested females spent longer durations in the intermediate area in photophase (11.9%) compared with scotophase (3.3%) in dsCON (see Table 2). It is also noteworthy that the tested females stayed a long time in the edge area (70.0~73.3%) compared with other micro-areas in both treatments (e.g., Table 1). This may indicate that the tendency to stay near boundary would persist after PKG modulation.

Durations (%) in the intermediate area were extremely high during P12~S2 compared with other light phases in dsPKG (see Figure 6B). However, no distinctive statistical differentiation was observed among light phases in duration (%) in the intermediate area in dsPKG (Figure 7). This would be due to the extremely high levels of SDs of durations (%) observed in this period, P12~S2 (Figure 6B). Further physiological and genetic investigations are required along with more trials to confirm if diel differences would exist in durations (%) in the intermediate area in dsPKG.

During P1, very high speed was observed in both treatments (see Figure 5A). This would suggest high activity after feeding in the previous stage in photophase (see Figure 4A), as discussed above. But the speed was also high in P1 for dsPKG, although feeding did not occur in the previous stage for dsPKG (see Figure 4B). This may be because the starting location for the test individuals was the edge area to secure minimum disturbance to test females in transferring them from the stock to the observation arena. Although the test females were acclimated for 2 h before observation (see Section 2.7. *Behavior-monitoring and automatic digitization*), they may have stayed longer and have been alerted to the new environment in the edge area in the initial phase of observation. More investigations are thus required in the future regarding examinations of mechanisms causing high speed in P1 in dsPKG or effect of acclimation on initial behavior in the observation arena.

In this study, measurements in different light phases were considered as independent for statistical analyses (see Section 2.8. Statistical analysis). When one-way repeated ANOVA was applied to dependent measurements of parameters and durations (%), statistical differences were not observed between light phases. This would be partly due to the low number of trials and partly due to the heterogenous conditions of the tested females. In the future, a more extensive experimental design will be needed including an increase in trial numbers and increase in physiological homogeneity of test females (e.g., precise range in age) as well. In this study, parameters in different micro-areas were not separately measured as time progressed. Some empirical results were presented including the speed change in the edge area (Figure 4). Future research will be required to quantitatively investigate the coupled dependence, micro-areas in space and light phases in time, throughout the observation period, possibly along with an increase in the number of trials.

With continuous and concurrent observation of movement activity and durations of stay in areas of the observation arena, the computational analysis of response behaviors supported physiological evidence of dsPKG suppression effects, demonstrating changes in behavioral status according to instantaneous movement parameters and durations (%) in different micro-areas, and revealing consecutive behavioral changes as well. Further study relating molecular and physiological mechanisms to behavior is warranted in the future to illustrate genetic functioning in an integrated manner.

## Figures and Tables

**Figure 1 insects-16-00320-f001:**
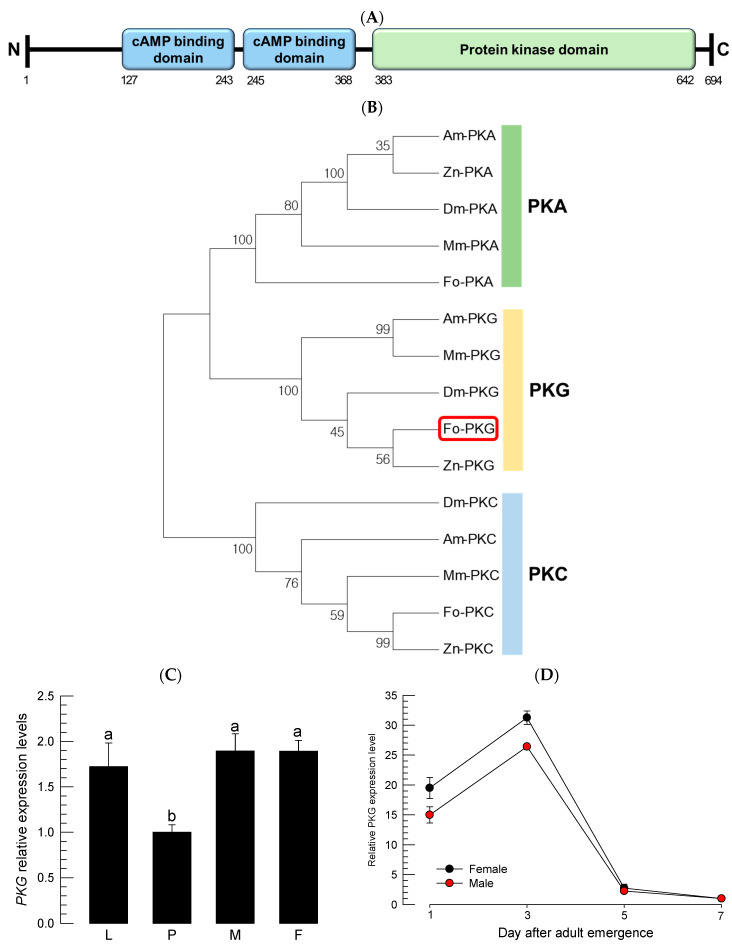
Molecular properties of *F. occidentalis* cGMP-dependent protein kinase (*Fo-PKG*). (**A**) Prediction of functional domains using Interpro (http://www.ebi.ac.uk/interpro/, accessed on 1 October 2024). Three domains are indicated on the amino acid sequence predicted from the open reading frame of *Fo-PKG*. (**B**) A phylogeny analysis of *Fo-PKG* with other PKG, cAMP-dependent protein kinase (*PKA*), and protein kinase C (*PKC*) genes using MEGA6. These genes are retrieved from GenBank with accession numbers of XP_052121837.1 for *Fo-PKA*, NP_476977.1 for *Drosophila melanogaster* PKA (*Dm-PKA*), XP_393285.1 for *Apis mellifera* PKA (*Am-PKA*), XP_021936510.1 for *Zootermopsis nevadensis* PKA (*Zn-PKA*), XP_057317931.1 for *Microplitis mediator* PKA (*Mm-PKA*), XM_026417160.2 for *Fo-PKG*, ACO44430.1 for *D. melanogaster* PKG (*Dm-PKG*), XP_026300309.1 for *A. mellifera* PKG (*Am-PKG*), XP_021933460.1 for *Z. nevadensis* PKG (*Zn-PKG*), XP_057321318.1 for *M. mediator* PKG (*Mm-PKG*), XP_052133408.1 for *F. occidentalis* PKC (*Fo-PKC*), NP_001287577.1 for *D. melanogaster* PKC (*Dm-PKC*), XP_026296808.1 for *A. mellifera* PKC (*Am-PKC*), XP_021920016.1 for *Z. nevadensis* PKC (*Zn-PKC*), and XP_057338986.1 for *M. mediator* PKC (*Mm-PKC*). (**C**) Expression profile of *Fo-PKG* in different developmental stages of larva (L), pupa (P), adult male (M), and adult female (F). (**D**) Change of expression levels of *Fo-PKG* in different adult ages. Expression level of an elongation factor, *Fo-EF1*, in each sample was used to normalize the expression level. In each treatment, three trials were used. Different letters above standard deviation bars indicate significant differences among means at Type I error = 0.05 (LSD test).

**Figure 2 insects-16-00320-f002:**
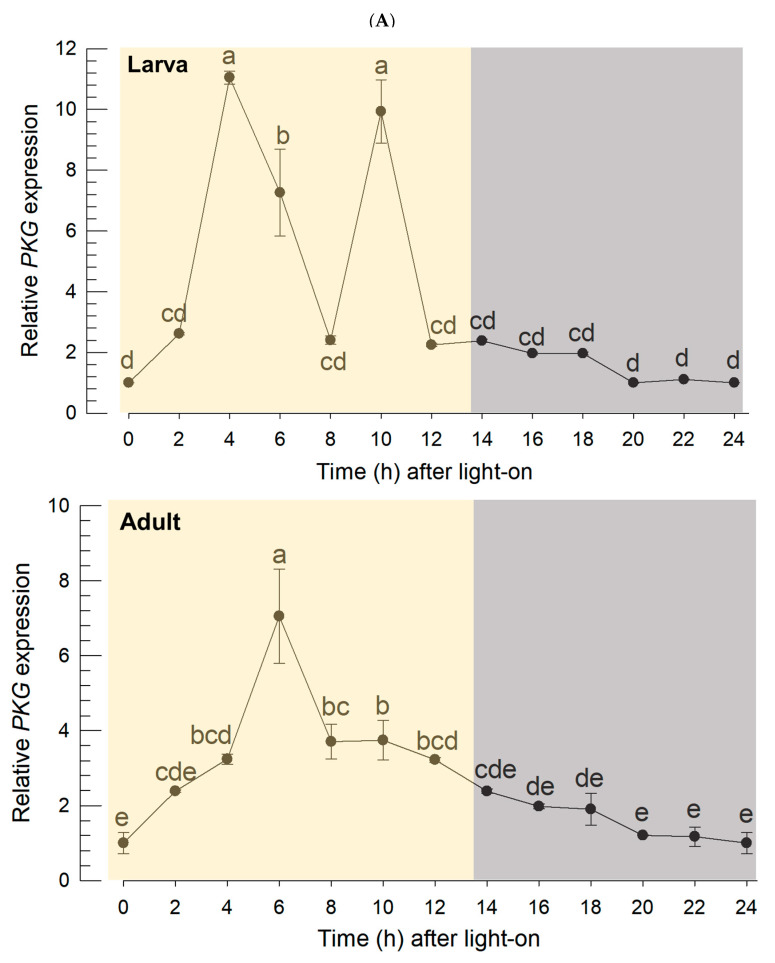
Diel rhythmicity of *PKG* expression in *F. occidentalis*. (**A**) RT-qPCR analysis of the gene every 2 h in larva (**upper** panel) and adult (**lower** panel) stages. Each measurement used 10 thrips and was independently repeated three times. Expression level of an elongation factor, *Fo-EF1*, was used to normalize the expression level. Different color backgrounds indicate photophase (0~14 h) and scotophase (14~24 h). (**B**) FISH analysis. Female adults were selected at photophase (6 h) and scotophase (18 h). Specific expression of *PKG* of *F. occidentalis* was observed with FITC-labeled antisense or sense probe (**upper** panel). A fluorescent microscope (DM2500; Leica, Wetzlar, Germany) was used to view the samples in fluorescence (‘FITC’ against the probe and ‘DAPI’ against nucleus) while the intact morph was visualized in a mode of differential interference contrast (DIC) at 100x magnification. The scale bar represents 0.1 mm. The intensity was quantified by normalizing the FITC and DAPI signals (**lower** panel). Each treatment was replicated three times and each trial contained three female adults. Different letters above the standard deviation bars indicate significant differences between means at Type I error = 0.05 (LSD test).

**Figure 3 insects-16-00320-f003:**
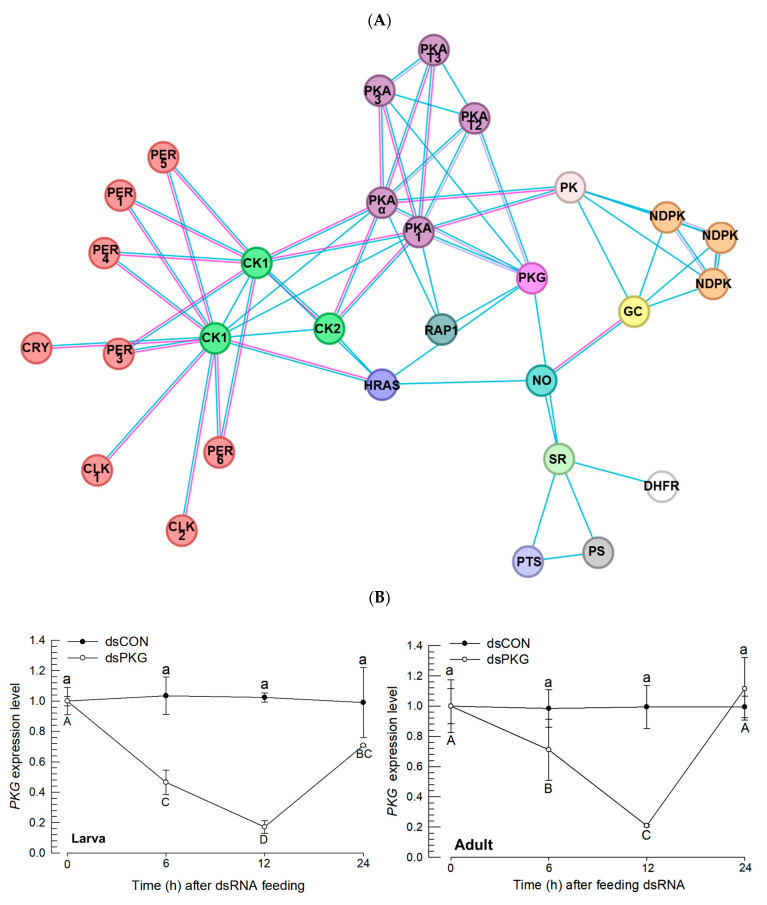
Functional association of PKG with clock genes in *F. occidentalis*. (**A**) A prediction of PKG interaction with other proteins: PKG (cGMP-dependent protein kinase), PER1 (Period circadian protein isoform X1), PER3 (Period circadian protein isoform X3), PER4 (Period circadian protein isoform X4), PER5 (Period circadian protein isoform X5), PER6 (Period circadian protein isoform X6), CLK1 (Circadian clock-dsCONled protein daywake 1), CLK2 (Circadian clock-dsCONled protein daywake 2), CRY (Cryptochrome-1), CK1 (Casein kinase I subunit beta), CK2 (Casein kinase II subunit alpha), HRAS (GTPase HRas), RAP1 (Rap1 GTPase-activating protein 1), PKAα (cAMP-dependent protein kinase catalytic subunit alpha), PKA1 (cAMP-dependent protein kinase catalytic subunit 1), PKA3 (cAMP-dependent protein kinase catalytic subunit 3), PKAT2 (cAMP-dependent protein kinase type II regulatory subunit), PKAT3 (cAMP-dependent protein kinase type III regulatory subunit), PK (pyruvate kinase), NDPK (nucleoside diphosphate kinase), GC (guanylate cyclase), NO (nitric oxide synthase), SR (sepiapterin reductase), DHFR (dihydrofolate reductase), PTS (6-pyruvoyltetrahydropterin synthase), and PS (phosphodiesterase). The map was generated using STRING 12.0 (https://string-db.org, assessed on 1 October 2024). (**B**) RNAi efficacy of dsRNA feeding against *PKG* expression in *F. occidentalis*. Following dsPKG treatment, the expression levels of the *Fo-PKG* gene were relatively assessed compared to those of the control (dsCON) at different time points with 10 thrips per measurement. The qPCR was independently repeated three times. For dsCON, a nontarget gene, *EGFP*, was used for RNAi dsCON (dsCON). An elongation factor, *Fo-EF1*, was used to normalize the expression level. Different letters above or below standard deviation bars indicate significant difference among means at Type error = 0.05 (LSD test) in each of dsCON and dsPKG. (**C**) Influence of RNAi specific to *PKG* expression on the expressions of clock genes: Clock (*Clk*) and Period (*Per*). After 24 h of dsPKG treatment, *CLK* and *Per* gene expressions were assessed using whole body samples at two time points in both photophase and scotophase.

**Figure 4 insects-16-00320-f004:**
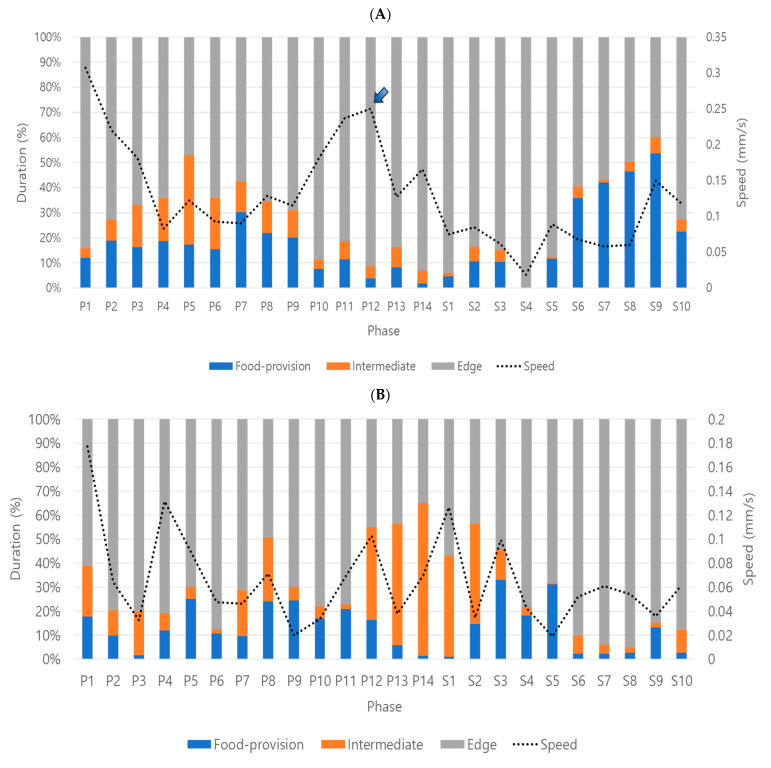
Durations (%) in micro-areas superimposed with speed of female thrips. (**A**) dsCON (four females) and (**B**) dsPKG (three females). Arrow indicates the highest speed.

**Figure 5 insects-16-00320-f005:**
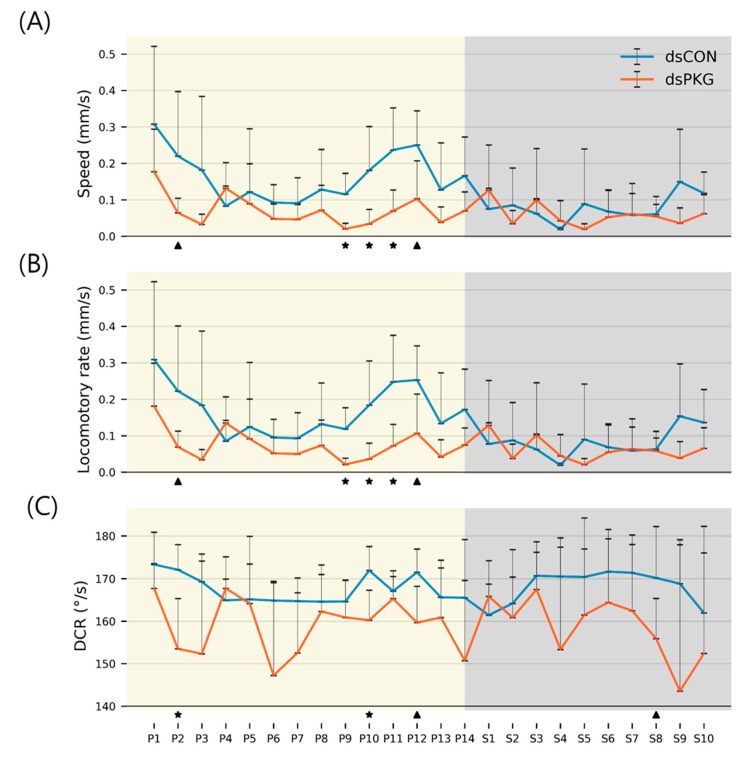
Movement parameters of female thrips. (**A**) Speed, (**B**) locomotory rate, (**C**) direction change rate (DCR) (‘⋆’ alpha error up to 0.05 and ‘▲’ alpha error up to 0.10 underneath each subfigure).

**Figure 6 insects-16-00320-f006:**
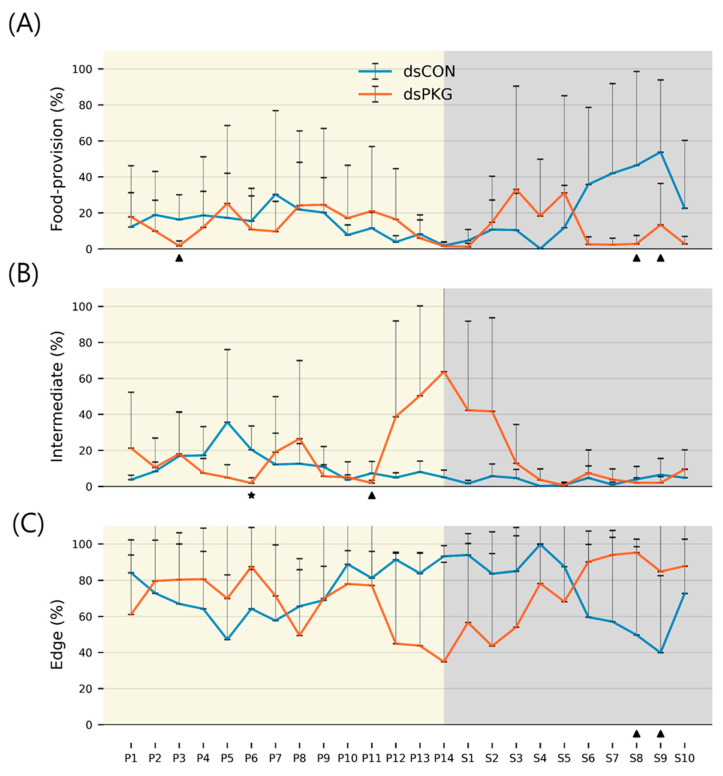
Durations (%) of female thrips staying in micro-areas. (**A**) Food-provision, (**B**) intermediate and (**C**) edge (‘⋆’ alpha error up to 0.05 and ‘▲’ alpha error up to 0.10 underneath each subfigure).

**Figure 7 insects-16-00320-f007:**
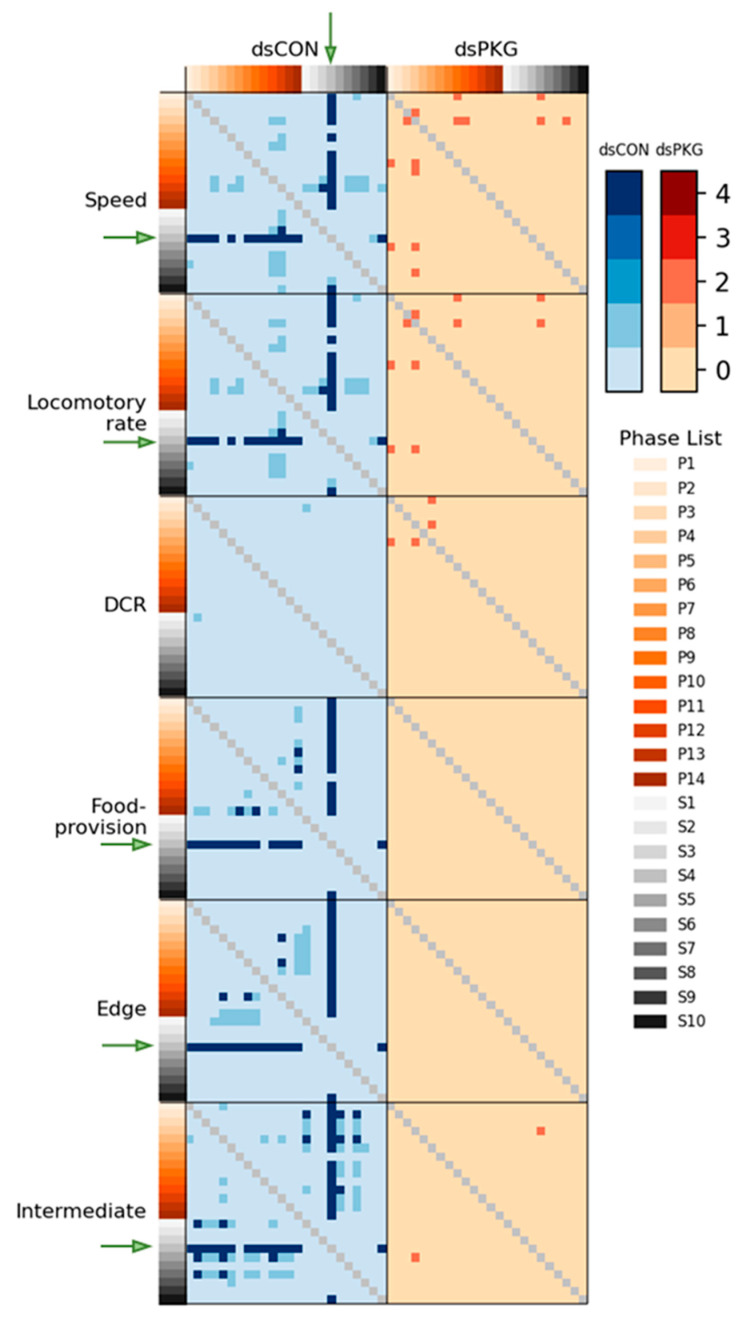
Statistical differentiation between dsCON and dsPKG in each hour phase in a 24 h cycle (14 h photophase (P1–P14) and 10 h scotophase (S1–S10)) according to combined results from Mann-Whitney U and Kolmogorov-Smirnov tests with higher shade presenting stronger statistical differentiation (see text), in which movement parameters and duration rates (%) in different micro-areas were compared among different light phases (see green arrows indicating S4 with significant difference from other light phases.).

**Table 1 insects-16-00320-t001:** Summary of movement parameters and durations (%) in different micro-areas in the observation arena for female thrips observed for 24 h continuously in dsCON and dsPKG.

	Treatment	dsCON		dsPKG	
Parameters		Mean	SD	Mean	SD
**Movement**	**Speed**	0.13 ± 0.12		0.07 ± 0.07	
	**Locomotory**	0.13 ± 0.12		0.07 ± 0.07	
	**DCR**	167.76 ± 7.87		158.93 ± 15.05	
**Duration (%)**	**Food-provision**	18.41 ± 27.43		13.30 ± 24.92	
	**Intermediate**	8.33 ± 13.20		16.68 ± 28.98	
	**Edge**	73.26 ± 29.59		70.02 ± 36.39	

**Table 2 insects-16-00320-t002:** Movement parameters and durations (%) in different micro-areas in the observation arena for female thrips in photo- and scotophases observed for 24 h continuously in dsCON and dsPKG.

	Treatment Light Phases	dsCON				dsPKG			
		Photophase		Scotophase		Photophase		Scotophase	
Parameters		Mean	SD	Mean	SD	Mean	SD	Mean	SD
**Movement**	**Speed**	0.16 ± 0.13		0.08 ± 0.08		0.07 ± 0.07		0.06 ± 0.07	
	**Locomotory**	0.17 ± 0.13		0.08 ± 0.09		0.07 ± 0.07		0.06 ± 0.07	
	**DCR**	167.51 ± 6.08		168.12 ± 9.83		159.09 ± 12.90		158.70 ± 17.58	
**Duration (%)**	**Food-provision**	14.55 ± 20.22		23.81 ± 34.41		14.09 ± 23.77		12.20 ± 26.40	
	**Intermediate**	11.89 ± 15.80		3.34 ± 5.08		19.62 ± 31.12		12.56 ± 25.10	
	**Edge**	73.56 ± 24.62		72.85 ± 35.40		66.29 ± 37.11		75.24 ± 34.68	

## Data Availability

Protein sequences are found in GenBank under the accession numbers given in the caption of Figure 1.

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
