# Peer review of "A Computational Analysis Based on Automatic Digitization of Movement Tracks Reveals the Altered Diurnal Behavior of the Western Flower Thrips, Frankliniella occidentalis, Suppressed in PKG Expression"

_insects, 2025, doi:10.3390/insects16030320_

Round 1

Reviewer 1 Report

Comments and Suggestions for Authors

Title: Constructed Excellently (If possible, to do short)

Introduction: The introduction is well-structured and effectively sets the stage for the content that follows.

Abstract: Well written.

Materials and Methods (M&M): The methodology section is exemplary, demonstrating a thorough and precise approach to the research process.

Results: The results section is flawless, with no errors or redundancies noted. The data presentation is clear and concise, ensuring the accuracy and reliability of the findings.

Discussion: The discussion is notably compelling, providing insightful analysis and vigorous interpretation of the results in relation to existing literature and future implications.

Comments on the Quality of English Language

English writing is good not required. 

Author Response

Comment #1: Introduction: The introduction is well-structured and effectively sets the stage for the content that follows. Abstract: Well written. Materials and Methods (M&M): The methodology section is exemplary, demonstrating a thorough and precise approach to the research process. Results: The results section is flawless, with no errors or redundancies noted. The data presentation is clear and concise, ensuring the accuracy and reliability of the findings.

Discussion: The discussion is notably compelling, providing insightful analysis and vigorous interpretation of the results in relation to existing literature and future implications.

Response: We appreciate your encouragements. In addition, we revised the manuscript with valuable comments raised by other reviewers. After all revision, the corresponding authors edited the entire manuscript to keep consistency.

Reviewer 2 Report

Comments and Suggestions for Authors

This is an impressive study. It uses sophisticated methods, takes an integrative approach from gene expression to behavior, provides important new data on the biology of a global crop pest, and demonstrates effective use of methods for expanding research on circadian rhythms from a small number of lab-based model organisms to field organisms and crop pests. This study also demonstrates the importance of taking time of day into account in any assays of gene expression. The experimental designs and statistical analyses are rigorous. 

There is one important theoretical issue that needs clarification, and some minor grammatical suggestions. 

  1. The one critical suggestion I have is to clarify the difference between analysis of a diel rhythm (e.g. activity and transcription in this case) and analysis of a circadian rhythm. The entire analysis was conducted on rhythms in a light:dark photoperiod, which means that the diel rhythms could be driven by, or altered by the photoperiod and not by a circadian clock mechanism. The most direct way to demonstrate that the diel rhythms are circadian is to demonstrate that they persist  under constant conditions, in the absence of the light:dark cycle.  Another way to demonstrate that the diel rhythms are driven by a circadian clock is to show that altering the circadian clock mechanism alters the rhythm: the authors do show that PKG RNAi alters the expression of two circadian clock genes and the activity rhythm, but these could be independent effects- it's not clear that the RNAi effect on the clock genes is causing the effect on the behavioral rhythm- that could be directly mediated through PKG expression.  This result does, however, demonstrate that PKG expression is regulating the expression of Per and Clock (assuming no collateral effects of the RNAi on other genes), and the authors show associated changes in both clock gene expression and the behavioral rhythm, which is consistent with circadian clock regulation of the behavior. I think the authors have clearly demonstrated the importance of PKG in regulating circadian activity rhythms here, but as they point out in the discussion, the mechanisms mediating clock, PKG, physiological mechanisms and behavioral mechanisms are not clear and require further study. The authors are in an excellent position to conduct further studies, and this research is an excellent and impressive demonstration that these mechanisms are functionally linked. So far, however, in the absence of direct manipulation of the circadian clock mechanism as an independent variable, or demonstration of a free-running circadian activity rhythm in the thrips, the authors need to be careful not to over-interpret their results with respect to whether the PKG effects on behavior are mediated through a circadian clock mechanism or some other mechanism, and whether the PKG effects on clock genes are causal or incidental with respect to the changes in behavior. Is PKG, for example, regulating a circadian clock which in turn alters the behavior, or is it coupling clock function to the behavior (like PDF in fruit flies), or both? 
  2. The only other non-grammatical point is that Figure 7 could be explained more clearly. Combining Kolmogorov-Smirnoff and Mann-Whitney statistical analysis is not a commonly used method, so the authors might include, in the Figure legend, a little more guidance on how one should interpret the figure. Also, the figure legend notes "A" and "B" type data, but there is no "A" or "B" on the figure itself so it's not clear which element in the graphs correspond to the A and B variables. The method itself is clever and well done but it's difficult to understand the figure as described. 
  3. Line 40: replace "the gene expression" with "PKG gene expression"?
  4. Lines 56-57: replace "under the oscillating expressions of the" with "by interacting with the expression of circadian"?
  5. Lines 61-62: replace "give a feeding" to "causes"?
  6. Line 63: replace "virus" with "viruses" and causing a devastating" to " causing devastating"?
  7. Lines 64-65: replace "hiding behavior into flowers" with "ability to hide in flowers and "to be effectively controlled by the" with effectively control"?
  8. Line 67: replace "suppress the outbreaks" with "suppress outbreaks"?
  9. Line 68: replace "to sticky trap" with "in sticky traps"?
  10. Line 69: replace "the visual" with "visual" and "useful" to "used" and "of the thrips under the" with and facilitated"?
  11. Line 78: replace "influence of" with "influences"?
  12. Line 79: delete "around diet"?
  13. Line 82: replace "depending on the onset of light signal by" with "by signals mediated through"?
  14. Line 83: replace "and catalyzes" with "which catalyzes"?
  15. Line 85: replace clock remained" with "clock function remained"?
  16. Line 88: delete "this study predicted", replace "PKG gene in" with "PKG gene expression in, and replace "and its expression levels were" with "was"?
  17. Line 89: replace "during 24-h period" with "over a 24-h period in a 12:12 LD cycle"?
  18. Line 92-93: replace "automatically detected data of the movement tracks by a continuous 24 h-monitoring device" with "continuous automated 24-hour monitoring of movement tracks"? 
  19. Line 524: replace "exhibited" with "showed" and "by" with "with"?
  20. Line 526: replace "kind of the" with "kind of" and replace "controlled by" with "consistent with" and replace "machinery equipped with" with "regulation mediated through"?
  21. Line 527: replace Per/TIM" with "the Per/TIM"?
  22. Line 536: replace "with the clock gene expressions" with "clock gene expression"?
  23. Line 555: replace "study" with "studies"?
  24. Line 612: replace "staying near" with "to stay near"?
  25. Lines 630-631: replace "on reasonable guidance for" with "relating" and replace "molecular physiological approach to behavioral data" with "molecular and physiological mechanisms to behavior""?
  26. Line 632: replace "in illustrating" with "to illustrate"? 

Comments on the Quality of English Language

Minor grammatical edits 

Author Response

Comment #2-1: This is an impressive study. It uses sophisticated methods, takes an integrative approach from gene expression to behavior, provides important new data on the biology of a global crop pest, and demonstrates effective use of methods for expanding research on circadian rhythms from a small number of lab-based model organisms to field organisms and crop pests. This study also demonstrates the importance of taking time of day into account in any assays of gene expression. The experimental designs and statistical analyses are rigorous.

There is one important theoretical issue that needs clarification, and some minor grammatical suggestions.

Response: We appreciate your clear understanding on the manuscript. Other comments are carefully reflected in the revised version.

Comment #2-2: The one critical suggestion I have is to clarify the difference between analysis of a diel rhythm (e.g. activity and transcription in this case) and analysis of a circadian rhythm. The entire analysis was conducted on rhythms in a light:dark photoperiod, which means that the diel rhythms could be driven by, or altered by the photoperiod and not by a circadian clock mechanism. The most direct way to demonstrate that the diel rhythms are circadian is to demonstrate that they persist under constant conditions, in the absence of the light:dark cycle.  Another way to demonstrate that the diel rhythms are driven by a circadian clock is to show that altering the circadian clock mechanism alters the rhythm: the authors do show that PKG RNAi alters the expression of two circadian clock genes and the activity rhythm, but these could be independent effects- it's not clear that the RNAi effect on the clock genes is causing the effect on the behavioral rhythm- that could be directly mediated through PKG expression.  This result does, however, demonstrate that PKG expression is regulating the expression of Per and Clock (assuming no collateral effects of the RNAi on other genes), and the authors show associated changes in both clock gene expression and the behavioral rhythm, which is consistent with circadian clock regulation of the behavior. I think the authors have clearly demonstrated the importance of PKG in regulating circadian activity rhythms here, but as they point out in the discussion, the mechanisms mediating clock, PKG, physiological mechanisms and behavioral mechanisms are not clear and require further study. The authors are in an excellent position to conduct further studies, and this research is an excellent and impressive demonstration that these mechanisms are functionally linked. So far, however, in the absence of direct manipulation of the circadian clock mechanism as an independent variable, or demonstration of a free-running circadian activity rhythm in the thrips, the authors need to be careful not to over-interpret their results with respect to whether the PKG effects on behavior are mediated through a circadian clock mechanism or some other mechanism, and whether the PKG effects on clock genes are causal or incidental with respect to the changes in behavior. Is PKG, for example, regulating a circadian clock which in turn alters the behavior, or is it coupling clock function to the behavior (like PDF in fruit flies), or both?

Response: This comment is helpful to explain the limitation of this study. Thus we rephrased the discussion related with this issue as follows: “The continuous monitoring of the thrips behavior showed a diurnal rhythmicity with relatively high activity during photophase and relatively low activity during scotophase. This kind of diel pattern is consistent with its circadian regulation mediated through two oscillatory loops: the Per/TIM oscillatory loop and the CRY loop [27,28]. In each of these loops, CLK/CYC acts as a transcriptional activator that promotes Per and TIM, or CRY transcription. The product proteins Per and TIM, or CRY are thought to provide negative feedback to inhibit the transcriptional activator. In D. melanogaster, CRY is expressed in specific clock neurons in the brain [29]. CRY is then activated by blue light and catalyzes TIM degradation through its protease activity, at which point the circadian clock is reset [30,31]. In F. occidentalis, Per and CLK genes exhibited the diel patterns with relatively high expressions during photophase and relatively low expressions during scotophase. This supports the circadian rhythmicity of the thrips behaviors controlled by clock genes be-cause any interruption of the clock gene expressions significantly altered the diel rhyth-micity [8]. Interestingly, this current study showed that PKG expression followed this diel pattern, suggesting a functional association of its expression with clock gene expression.

RNAi specific to PKG expression led to alterations in the clock gene expressions dur-ing photophase by preventing up-regulation of the clock genes. PKG expression is associ-ated with division of labor in honey bee, Apis mellifera, in which specific expression in the mushroom bodies of the brain, suboesophageal ganglion, and corpora allata is associated with foraging behavior to collect pollens and nectars [9]. The role of PKG expression in the labor division in social insects was also found in a fire ant, Solenopsis invicta, in which RNAi specific to PKG reduced the locomotory activity and facilitated the behavioral change from foragers to nurses [32]. PKG activity and locomotory activity has been well established in Drosophila, in which flies are discriminated into sitters with low PKG activ-ity or rovers with high PKG activity [33]. These suggest that PKG expression is associated with high locomotory activities of F. occidentalis during photophase. Thus, the RNAi spe-cific to PKG expression resulted in reduced immature development and adult fecundity presumably by inhibiting feeding and ovipositional behavior. In addition, the suppression of PKG expression led to the suppression of the clock gene expressions during photophase, supporting the functional association of the PKG expression with the circadian rhythmic-ity. In mammals, the suppression of clock gene expressions led to the suppression of PKG protein level [34]. This suggests that the alterations of the clock gene expressions may be indirect influence of the reduced PKG mRNA level caused by RNAi in F. occidentalis. The control of PKG level by the clock gene expressions needs to be addressed in future studies. It is also noteworthy that there are at least four isoforms of PKG in F. occidentalis genome: 176 (XP_052119469.1), 417 (XP_052132331.1), 694 (XP_026272945.1), and 1,010 (XP_026272942.1) amino acid residues. In this study, we analyzed only PKG with 694 amino acid length isoform. Thus, the other isoforms should be analyzed in their inde-pendent roles. In addition, our current study showed the altering expressions of two cir-cadian genes by manipulating PKG expression but did not show the control of the PKG gene expression by the circadian gene expression. Thus, it is still unclear whether the PKG effects on clock genes are causal or incidental with respect to the changes in behavior. This needs to be clarified in subsequent study.”

Comment #2-3: The only other non-grammatical point is that Figure 7 could be explained more clearly. Combining Kolmogorov-Smirnoff and Mann-Whitney statistical analysis is not a commonly used method, so the authors might include, in the Figure legend, a little more guidance on how one should interpret the figure. Also, the figure legend notes "A" and "B" type data, but there is no "A" or "B" on the figure itself so it's not clear which element in the graphs correspond to the A and B variables. The method itself is clever and well done but it's difficult to understand the figure as described.

Response: Explanations on interpreting statistical results are added in the legend of Figure 7, and ‘(A) and (B)’ in Figure 7 are eliminated, as advised. Also additional statements regarding assumptions of statistical differences are provided in ‘Materials and Methods’ and ‘Discussion’: [Figure 7] ‘Statistical differentiation between dsCON and dsPKG in each hour phase in 24-h cycle (14-hour photophase (P1-P14) and 10-hour scotophase (S1-S10)) according to combined results from Mann-Whitney U and Kolmogorov-Smirnov tests with higher shade presenting stronger statistical differentiation (see text), in which movement parameters and duration rates (%) in different micro-areas were compared among different light phases (see green arrows indicating S4 with significant difference from other light phases.)’ [Materials and Methods] ‘Two aspects of statistical difference were considered in this study: difference between dsCON and dsPKG at each light phase, and difference among light phases within each strain. Behavior data in this study had a property of measurement dependence. Since observations were continuously conducted throughout the whole observation period, movement across light phases are dependently observed (i.e., observed continuously). In this study measurements in different light phases were considered as independent since this study is an initial phase of behavior study to confirm physiological effects, and an extensive experimental design would be required for analyzing repeated measurements possibly with large number of trials.’ [Discussion] ‘In this study measurements in different light phases were considered as independent for statistical analyses (see Section 2.8. Statistical analysis). When one-way re-peated ANOVA was applied to dependent measurements of parameters and durations (%), statistical differences were not obtained between light phases. This would be partly due to low number of trials and partly due to heterogenous conditions of tested females. In the future a more extensive experimental design will be needed including an increase in trial numbers and increase in physiological homogeneity of test females (e.g., precise range in age) as well.’

Comment #2-4: Line 40: replace "the gene expression" with "PKG gene expression"?

Response: Corrected as suggested

Comment #2-5: Lines 56-57: replace "under the oscillating expressions of the" with "by interacting with the expression of circadian"?

Response: Corrected as suggested

Comment #2-6: Lines 61-62: replace "give a feeding" to "causes"?

Response: Corrected as suggested

Comment #2-7: Line 63: replace "virus" with "viruses" and causing a devastating" to " causing devastating"?

Response: Corrected as suggested

Comment #2-8: Lines 64-65: replace "hiding behavior into flowers" with "ability to hide in flowers and "to be effectively controlled by the" with effectively control"?

Response: Corrected as suggested

Comment #2-9: Line 67: replace "suppress the outbreaks" with "suppress outbreaks"?

Response: Corrected as suggested

Comment #2-10: Line 68: replace "to sticky trap" with "in sticky traps"?

Response: Corrected as suggested

Comment #2-11: Line 69: replace "the visual" with "visual" and "useful" to "used" and "of the thrips under the" with and facilitated"?

Response: Corrected as suggested

Comment #2-12: Line 78: replace "influence of" with "influences"?

Response: Corrected as suggested

Comment #2-13: Line 79: delete "around diet"?

Response: Corrected as suggested

Comment #2-14: Line 82: replace "depending on the onset of light signal by" with "by signals mediated through"?

Response: Corrected as suggested

Comment #2-15: Line 83: replace "and catalyzes" with "which catalyzes"?

Response: Corrected as suggested

Comment #2-16: Line 85: replace clock remained" with "clock function remained"?

Response: Corrected as suggested

Comment #2-17: Line 88: delete "this study predicted", replace "PKG gene in" with "PKG gene expression in, and replace "and its expression levels were" with "was"?

Response: Corrected as suggested

Comment #2-18: Line 89: replace "during 24-h period" with "over a 24-h period in a 12:12 LD cycle"?

Response: Corrected as suggested

Comment #2-19: Line 92-93: replace "automatically detected data of the movement tracks by a continuous 24 h-monitoring device" with "continuous automated 24-hour monitoring of movement tracks"?

Response: Corrected as suggested

Comment #2-20: Line 524: replace "exhibited" with "showed" and "by" with "with"?

Response: Corrected as suggested

Comment #2-21: Line 526: replace "kind of the" with "kind of" and replace "controlled by" with "consistent with" and replace "machinery equipped with" with "regulation mediated through"?

Response: Corrected as suggested

Comment #2-22: Line 527: replace Per/TIM" with "the Per/TIM"?

Response: Corrected as suggested

Comment #2-23: Line 536: replace "with the clock gene expressions" with "clock gene expression"?

Response: Corrected as suggested

Comment #2-24: Line 555: replace "study" with "studies"?

Response: Corrected as suggested

Comment #2-25: Line 612: replace "staying near" with "to stay near"?

Response: Corrected as suggested

Comment #2-26: Lines 630-631: replace "on reasonable guidance for" with "relating" and replace "molecular physiological approach to behavioral data" with "molecular and physiological mechanisms to behavior""?

Response: Corrected as suggested

Comment #2-27: Line 632: replace "in illustrating" with "to illustrate"?

Response: Corrected as suggested

Reviewer 3 Report

Comments and Suggestions for Authors

The paper, titled “A computational analysis based on automatic digitization of movement tracks reveals the altered diurnal behavior of the western flower thrips, Frankliniella occidentalis, suppressed in PKG expression.” by Xia et al investigate the diurnal behavior of the western flower thrips is controlled by the circadian clock machinery. Authors show that a foraging gene encoding cGMP-dependent protein kinase (PKG) mediates this diurnal behaviors. Functional tests using RNA interference show significant alteration of clock gene expression and diel difference with speed and mobility of the animal. The paper is well written and data is clearly presented and statistically tested. I have a few concerns that are listed below.

Major comments:

  1. I did not understand why the authors tested diel rhythmicity of Fo-PKG expression only in the midgut region and not in the brain. Logically, most behaviors originate in the brain and are transmitted to different parts of the body. This has not been specified or discussed in the manuscript.
  2. In the results section 3.3, the authors assess the expression of Clk and Per; however, they do not mention the tissue used for RNA isolation.
  3. In Figure 3B, the authors compare PKG gene expression in control (dsCon) and knockdown (dsPKG) groups. The figure shows a flat line for control gene expression (dsCon), although the authors previously demonstrated that PKG expression is dynamic (Figure 2A). When was the expression tested?
  4. "In the results section 3.4, please briefly explain the behavior setup and cite the supplementary figure before presenting the results. In addition, please explain the rationale for defining specific areas for edge and intermediate regions? How is area segregation significant for behavior and circadian rhythm?"
  5. The results section 3.4 is very long, spanning 6-7 pages and including four figures and two tables. This section is difficult to follow in terms of logic, results, and conclusions, which may hinder readers' appreciation of the work's importance. I suggest the authors break this section into at least two parts.
  6. In the discussion section, the authors mention that- 'Thus, the RNAi specific to PKG expression resulted in reduced immature development and adult fecundity, presumably by inhibiting feeding and ovipositional behavior.' Please cite a paper that suggests poor feeding causes immature development and reduced fecundity.
  7. In the discussion section, the authors state, 'The control of PKG level by clock gene expressions needs to be addressed in future studies.' This statement is incorrect because the data suggest that PKG controls the expression of clock genes, not the other way around. This statement should be either removed or restructured.

Minor comments:

  1. Both sections (A and B) of Figure 2 contain two figures. The upper and lower figures in each section are not specified in the results or figure legends. It may help readers if these can be numbered separately.
  2. Please redefine DCR (Direction Change Rate) in the results section 3.4. Most readers refer to the results before consulting the materials and methods.
  3. Please mention how many animals were tested in Figure 4.

Author Response

Comment #3-1: The paper, titled “A computational analysis based on automatic digitization of movement tracks reveals the altered diurnal behavior of the western flower thrips, Frankliniella occidentalis, suppressed in PKG expression.” by Xia et al investigate the diurnal behavior of the western flower thrips is controlled by the circadian clock machinery. Authors show that a foraging gene encoding cGMP-dependent protein kinase (PKG) mediates this diurnal behaviors. Functional tests using RNA interference show significant alteration of clock gene expression and diel difference with speed and mobility of the animal. The paper is well written and data is clearly presented and statistically tested. I have a few concerns that are listed below.

Response: We appreciate your clear understanding on the manuscript. Other comments are carefully reflected in the revised version.

Comment #3-2: I did not understand why the authors tested diel rhythmicity of Fo-PKG expression only in the midgut region and not in the brain. Logically, most behaviors originate in the brain and are transmitted to different parts of the body. This has not been specified or discussed in the manuscript.

Response: It is a logical comment. This study focused on the feeding behavior to assess the circadian rhythm. The diel pattern of the feeding behavior is controlled by central (brain) and peripheral (fat body and gut) signals because any mismatch between these signals would lead to significant decease of feeding activity in Drosophila (Fulgham et al., 2021). 

Fulgham CV, Dreyer AP, Nasseri A, Miller AN, Love J, Martin MM, Jabr DA, Saurabh S, Cavanaugh DJ. Central and Peripheral Clock Control of Circadian Feeding Rhythms. J Biol Rhythms. 2021 Dec;36(6):548-566.  

Comment #3-3: In the results section 3.3, the authors assess the expression of Clk and Per; however, they do not mention the tissue used for RNA isolation.

Response: We used whole body samples. The figure caption is rephrased as follows: “After 24 h of dsPKG treatment, CLK and Per gene expressions were assessed using whole body samples at two-time points in each of photophase and scotophase.”

Comment #3-4: In Figure 3B, the authors compare PKG gene expression in control (dsCon) and knockdown (dsPKG) groups. The figure shows a flat line for control gene expression (dsCon), although the authors previously demonstrated that PKG expression is dynamic (Figure 2A). When was the expression tested?

Response: They are relative expression levels compared to controls. We added this information to the caption as follows: “Following dsPKG treatment, the expression levels of the Fo-PKG gene were relatively assessed compared to those of the control (dsCON) at different time points with 10 thrips per measurement.”

Comment #3-5: "In the results section 3.4, please briefly explain the behavior setup and cite the supplementary figure before presenting the results. In addition, please explain the rationale for defining specific areas for edge and intermediate regions? How is area segregation significant for behavior and circadian rhythm?"

Response: Explanations on behavior setup and definition of the edge area are provided in ‘Section 3.4 in Results’ and ‘Section 2.7 in Materials and Methods.’ Additionally, behaviors according to micro-areas are added in ‘Section 3.4 in Results’ and ‘Discussion’: [Results, Section 3.4. Parameter extraction and behavior profiles of adult females)] ‘From the digitized movement data, parameters including speed, locomotory rate and direction change rate (DCR) were extracted throughout the whole observation period (1 day). Concurrently durations (%) were measured in different micro-areas including the food-provision, intermediate and edge areas as time progressed (See Fig. S1).’ [Materials and Methods, Section 2.7. Behavior-monitoring and automatic digitization)] ‘According to measurements of 20 adult females, the length, width and height were 1.64 ± 0.07 mm, 0.29 ± 0.03 mm and 0.25 ± 0.01 mm, respectively. Considering the tested fe-males moved around about three times of either width or height of their bodies along the edge area according to preliminary experiments, 1 mm was determined as the width of the edge area. The intermediate area was defined as the area between the food-provision area and the edge area.’ [Results, Section 3.4. Parameter extraction and behavior profiles of adult females)] ‘It is also noted that speed was differentiated according to micro-areas across light phases. The females mostly stayed in the edge area (87.1 ± 9.5 %) during P10 – P12. Speed had was correspondingly very high (0.22 ± 0.10 mm/s) during this period with a peak during P12 (Fig. 4A). Moreover, the speed decreased to a low level (0.09 ± 0.09 mm/s) afterward during P13 – S6, while the duration in the edge area was still high with 85.8 ± 21.5 %. This indicated that speed changed within the edge area as time progressed. In the early photophase, the decrease in speed was similarly observed. The speed was initially in the highest level at 0.31 ± 0.19 mm/s during P1 and rapidly de-creased in the following light phases, P2, P3 and P4, with 0.22 ± 0.15 mm/s, 0.18± 0.18 mm/s and 0.08 ± 0.05 mm/s, respectively. In this period, the duration rates in the edge area also decreased, 84.1 ± 15.7 %, 72.8 ± 25.4 %, 66.9 ± 28.7 %, 64.2 ± 27.5 %, respectively (Fig. 4A). During the period S6 – S9, duration in the food provision area increased to 44.5 ± 40.8 %, somewhat like duration in the edge area (51.5 ± 40.4 %). During this period, the speed was in the intermediate range, 0.08 ± 0.09 mm/s. The results overall indicated that speed was variable according to micro-areas as time progressed, especially in the edge area for dsCON.’ and ‘Speed was not characterized according to different micro areas overall for dsPKG, since the parameters and durations (%) were irregular in this strain. Although irregular overall, it is noted that a sharp decrease in speed was observed from 0.18 mm/s to 0.03 mm/s along with the decrease in durations (%) in the edge area during P1 ~ P3 (Figure 4B), like the case shown for dsCON (Figure 4A). Also, long durations (%) in the intermediate area were observed from P12 to S2 (38.7 % ~ 63.7 %), while the speed was in the middle range (0.03 ~ 0.10 mm/s) during this period.’ [Discussion] ‘During P1, very high speed was observed in both treatments (see Figure 5A). This would present high activity after feeding in the previous stage in photophase (see Fig-ure 4A) as discussed above. But the speed was also high in P1 for dsPKG, although feeding did not occur in the previous stage for dsPKG (see Figure 4B). This may be because the starting location for the test individuals was the edge area to secure mini-mum disturbance to test females in transferring them from the stock to the observation arena. Although the test females were acclimated for 2 h before observation (see Sec-tion 2.7. Behavior-monitoring and automatic digitization), they may have stayed longer and have been alerted to new environment in the edge area in the initial phase of observation. More investigations are thus required in the future regarding examinations of mechanisms causing high speed in P1 in dsPKG or effect of acclimation to initial be-havior in the observation arena.’,,, ‘Future research will be required to quantitatively investigate the coupled dependence, micro-areas in space and light phases in time, throughout the observation period possibly along with an increase in the number of trials.’

Comment #3-6: The results section 3.4 is very long, spanning 6-7 pages and including four figures and two tables. This section is difficult to follow in terms of logic, results, and conclusions, which may hinder readers' appreciation of the work's importance. I suggest the authors break this section into at least two parts.

Response: As advised, section 3.4 is divided into ‘3.4. Parameter extraction and behavior profiles of adult females’ and ‘3.5. Comparison of movement trends between dsCON and dsPKG’ in revision.

Comment #3-7: In the discussion section, the authors mention that- 'Thus, the RNAi specific to PKG expression resulted in reduced immature development and adult fecundity, presumably by inhibiting feeding and ovipositional behavior.' Please cite a paper that suggests poor feeding causes immature development and reduced fecundity.

Response: It is provided in Supplementary Information in Fig. S2. This is newly added in this manuscript.

Comment #3-8: In the discussion section, the authors state, 'The control of PKG level by clock gene expressions needs to be addressed in future studies.' This statement is incorrect because the data suggest that PKG controls the expression of clock genes, not the other way around. This statement should be either removed or restructured.

Response: The statement is deleted.

Comment #3-9: Both sections (A and B) of Figure 2 contain two figures. The upper and lower figures in each section are not specified in the results or figure legends. It may help readers if these can be numbered separately.

Response: Added as follows: “Figure 2. Diel rhythmicity of PKG expression in F. occidentalis. (A) RT-qPCR analysis of the gene every 2 h in larva (upper panel) and adult (lower panel) stages. ……………… (B) FISH analysis. Female adults were selected at photophase (6 h) and scotophase (18 h). Specific expression of PKG of F. occidentalis was observed with FITC-labeled antisense or sense probe (upper panel). A fluorescent microscope (DM2500; Leica, Wetzlar, Ger-many) was used to view the samples in fluorescence (‘FITC’ against the probe and ‘DAPI’ against nucleus) while the intact morph was visualized in a mode of differential interference contrast (DIC) at 100x magnification. The scale bar represents 0.1 mm. The intensity was quantified by normal-izing the FITC and DAPI signals (lower panel). …………….”

Comment #3-10: Please redefine DCR (Direction Change Rate) in the results section 3.4. Most readers refer to the results before consulting the materials and methods.

Response: Direction change rate (DCR) is redefined in section 3.4 as advised: [Section 3.4.] ‘From the digitized movement data, parameters including speed, locomotory rate and Direction change rate (DCR) were extracted throughout the whole observation period (1 day).’

Comment #3-11: Please mention how many animals were tested in Figure 4.

Response: The number of tested females is listed in the Figure legend: [Figure 4] ‘Durations (%) in micro-areas superimposed with speed of female thrips. (A) dsCON (four females) and (B) dsPKG (three females).’

Round 2

Reviewer 3 Report

Comments and Suggestions for Authors

It looks much better now! Congratulations to the authors!